# Seroprevalence and Shifting Endemicities of Hepatitis A Virus Infection in Two Contrasting Geographical Areas in Indonesia

**DOI:** 10.3390/medicina61050806

**Published:** 2025-04-26

**Authors:** Dwi Prasetyo, Yudith Setiati Ermaya, Gustavo Hernandez-Suarez, Adriana Guzman-Holst, Cissy B. Kartasasmita

**Affiliations:** 1Department of Child Health, Faculty of Medicine, Padjadjaran University, Dr. Hasan Sadikin General Hospital, Bandung 40161, West Java, Indonesia; dwi.prasetyo@unpad.ac.id (D.P.); cbkarta@gmail.com (C.B.K.); 2Epidemiology & HEOR, Emerging Markets, GSK Vaccines, 1300 Wavre, Belgium; gustavo.x.hernandez@gsk.com (G.H.-S.); adriana.x.guzman@gsk.com (A.G.-H.)

**Keywords:** seroprevalence, endemicity, hepatitis A, age at midpoint of population immunity (AMPI)

## Abstract

*Background and Objectives:* Hepatitis A is an infectious disease caused by the hepatitis A virus (HAV), which is transmitted via the fecal–oral route, either through the consumption of contaminated food and water or through direct contact with an infected individual. The incidence of HAV is closely associated with socioeconomic factors, access to clean drinking water, sanitation safety, and hygiene. This study aimed to determine HAV seroprevalence and shifting endemicities of hepatitis A virus infection. The seroprevalence and endemicity status were assessed based on the age at the midpoint of population immunity (AMPI). *Materials and Methods:* A cross-sectional seroprevalence study was conducted in two contrasting areas (urban vs. rural) in Bandung, Indonesia. All participants underwent serological testing for anti-HAV IgG using a chemiluminescent microparticle immunoassay (CMIA) and participated in questionnaire interviews. Socioeconomic status was assessed using the Water/sanitation, Assets, Maternal education, and Income (WAMI) index. All statistical analyses were performed using SPSS 18, with a *p*-value of <0.05 considered significant. *Results:* A total of 1280 participants were tested (640 living in urban areas; 640 living in rural areas). The total prevalence of HAV seropositivity was 50.5% (95% confidence interval [CI]: 47.7–53.3%), with prevalences of 46.1% (95% CI: 42.5–54.4%) across urban sites and 54.7% (95% CI: 50.7–58.6%) across rural sites. The AMPI was within the 20–24-year age group, with an age point of 22 years, classified as an intermediate HAV endemicity status. *Conclusions:* the study found a shift in HAV endemicity status from low to intermediate, supporting the need for large-scale national hepatitis A vaccination in Indonesia.

## 1. Introduction

Hepatitis A is an infectious disease caused by the hepatitis A virus (HAV), transmitted predominantly via the fecal–oral route, either through the ingestion of contaminated water or food or by direct contact with an infectious person. The incidence of HAV is closely related to socioeconomic factors, with risk factors for HAV infection mainly relating to limited access to clean water and inadequate sanitation [1,2]. The World Health Organization (WHO) estimated an increase in the number of acute hepatitis A cases from 117 million in 1990 to 126 million in 2005, primarily in the 2–14-year and >30-year age groups [3,4]. Global Disease Burden data in 2019 reported an incidence of 159 million acute hepatitis A cases and 39,300 deaths attributed to the disease [5].

The body produces Immunoglobulin M (IgM) antibodies when first exposed to hepatitis A. These antibodies remain in the blood for about 2–12 months. Immunoglobulin G (IgG) antibodies appear after the body is infected with the hepatitis A virus. These antibodies appear 8 to 12 weeks after infection or vaccination, and they remain in the blood and protect against hepatitis A permanently [6].

Estimating the age-specific seroprevalence of anti-HAV IgG antibodies can indirectly show the epidemiological situation and endemicity levels of HAV in a country [7]. Although anti-HAV seroprevalence data for low- and middle-income countries are limited, a meta-analysis of published data conducted in 2005 found low to intermediate levels of endemicity in middle-income countries in Asia, the Middle East, Eastern Europe, and Latin America, as well as in Southeast Asian countries, such as Thailand, Indonesia, and Singapore [8]. Limitations in terms of data availability can make it difficult to identify shifts in endemicity [9]. The most recent HAV seroprevalence data in Indonesia indicated low endemicity, with an HAV seroprevalence of 28.65% in Yogyakarta [9,10].

Data regarding the differences in the seroprevalence of HAV between urban and rural areas in Southeast Asia are very limited. Therefore, when considering the overall endemicity within a country, it is important to measure HAV [11] seroprevalence separately in both urban and rural areas. Two contrasting geographical areas in Indonesia were studied: urban areas (lowlands), which has an altitude of 675–1050 m above sea level (asl), and the research location is around 768 m asl, with an average temperature of 23.5 °C, an average rainfall of around 200.4 mm per month, and vegetation varying from low to high greenery, and rural areas (highlands), which have an altitude ranging from 750 to 1200 m above sea level (asl), with an average monthly temperature of 23.1 °C, rainfall of around 250 mm per month, and high-density vegetation [12,13].

Hepatitis A vaccinations can reduce the incidence of hepatitis A in Indonesia; a hepatitis A vaccination program is available but is still limited to private services.

The primary objective of this study was to determine the current HAV age-specific seroprevalence of two contrasting areas (urban vs. rural) in Indonesia, and the secondary objective was to describe HAV endemicity by estimating the age at the midpoint of population immunity (AMPI). These data are important for policy makers in determining strategies to reduce HAV infections and disease burden, with the goal of eliminating hepatitis A from Indonesia by 2030 [14,15].

## 2. Materials and Methods

### 2.1. Study Design and Setting

An observational, cross-sectional seroprevalence study was conducted in two contrasting geographical areas in Bandung City (urban) vs. Bandung District (rural), West Java, Indonesia. The participants were between 1 and 80 years of age, had lived in the selected geographical area during the 6 months prior to the study, and provided informed consent or attestation by a parent/legal guardian for minors, according to local good research practices. A population-based, random, age-stratified sampling approach was used: participants were randomly selected from lists of residents provided by local health centers/authorities; if reliable lists could not be generated or had insufficient coverage (<85%), institution-based sampling was used instead. In this method, individuals attending primary care facilities were invited to participate. The sample size was calculated based on the age-specific seroprevalence data from Thailand in 1990 and 2004, giving a required sample size of 640 participants at each geographic location (see Appendix A for sample size estimation details and age stratification groups) [16].

### 2.2. Anti-HAV IgG Testing and Endemicity Measure

All recruited participants underwent anti-HAV IgG serological blood testing and questionnaire interviews. Testing for anti-HAV IgG antibodies was carried out by laboratory personnel using the FDA-approved ARCHITECT HAVAB-G two-step chemiluminescent immunoassay (CMIA; Abbott Laboratories) [17]. Specimens with signal-to-cut-off (S/CO) values of ≥1.00 were considered reactive (positive) for anti-HAV IgG; specimens with S/CO values of <1.00 were considered nonreactive (negative). This assay does not detect the IgM antibody (indicative of acute hepatitis); therefore, a positive result indicates a prior infection with HAV [18].

Endemicity was assessed using the AMPI, defined as the youngest age at which at least half of the population has serologic evidence of prior HAV infection; this was calculated using the equation of the best-fit curve for the youngest age at which the seroprevalence was equal to 50% [7]. Endemicity levels according to the AMPI are classified as follows: AMPI < 5 years = very high; AMPI 5–14 years = high, AMPI 15–34 years = intermediate, AMPI ≥ 35 years = low [7].

### 2.3. Measuring Risk Factors and Associations with HAV Seroprevalence

Predictors of past or recent infection with HAV were assessed with an interviewer-administered questionnaire built for the purposes of this study Appendix A [2,16,19,20]. The survey contained 44 questions divided into 5 sections: (a) sociodemographic (8 items), (b) knowledge of hepatitis A disease (7 items), (c) past medical history of hepatitis (10 items), (d) water safety access (9 items), and (e) hygienic food intake practices (10 items). The information collected included the attained education level (or education level of guardians for participants aged <18 years), household income and size, water sources, food and water usage practices, and access to sanitation facilities. These data were used to calculate socioeconomic status (SES) using the Water/sanitation, Assets, Maternal education and Income (WAMI) index Appendix A [21]. The WAMI index ranges from 0 to 1, with a higher value indicating a higher SES for the household.

### 2.4. Data Management and Analysis

Data are expressed as numbers and percentages for categorical data and expressed as means, standard deviations (SDs), medians, and ranges for numerical data. The statistical tests used to describe the strength of association (prevalence ratio [PR]) of the characteristics and risk factors of HAV infection were the Chi-square test, the Mann–Whitney test for comparing two medians, and logistic multiple regression for multivariate analysis. The multiple logistic regression method estimated the Nagelkerke R-Squared (R^2^) or the coefficient of determination, which measures the proportion of variance in a dependent variable that can be explained by the independent variable. All statistical analyses were performed using SPSS 18, with a *p*-value of <0.05 considered significant.

## 3. Results

### 3.1. Participant Characteristics

A total of 1280 participants were recruited, with 640 in urban areas and 640 in rural areas; 60.5% of the participants were female.

The most common education level of the participants’ mothers and fathers was high school (36.6% and 41.6%, respectively), the most common occupation (except for “attending school”) was household duties (24.9%), and most households (53.4%) had 4–5 family members living together. There were significant differences between seropositive and seronegative participants regarding the highest education levels of both fathers and mothers, the participants’ occupations, and the total number of family members living in the household.

Overall, there was higher seropositivity in females than in males (54.3% vs. 44.9%; *p* = 0.001) (Table 1). Only eight participants (four in each region, 0.6%) reported that they or their child had ever been diagnosed with hepatitis disease, three of which reported hepatitis A (in the other five cases, the type of hepatitis was unknown).

### 3.2. Seroprevalence and Endemicity

Hepatitis A seroprevalence was 50.6% (95% CI: 47.8–53.3%) overall, 46.4% (95% CI: 42.5–54.4%) in urban areas, and 54.7% (95% CI: 50.7–58.6%) in rural areas (Table 2).

The seropositive of hepatitis A was found to increase with age (Table 3).

Significant differences were seen in seropositivity in the 10–14-year age group (urban: 14.0%, rural: 44.0%; *p* = 0.001). The PRs of seropositivity for ages 3–4 and 5–9 were not significantly different between the urban and rural areas (Figure 1).

The best-fit curve for the percentage of seroprevalence versus age showed that the youngest age at which the seroprevalence was equal to 50% (AMPI) was in the 20–24 age group in the urban area, with an age point of 24.5 years, and in the 15–19 age group in the rural area, with an age point of 19.5 years. The overall AMPI showed that the seroprevalence was equal to 50% in the 20–24 age group (95% CI: 41.80–57.55%), with an age point of 22 years (Figure 2), corresponding to an “intermediate” level of endemicity (AMPI of 15–34 years) (Figure 3).

### 3.3. Household SES

The household SES was assessed using the WAMI index. The mean WAMI index was higher in urban areas (0.555, SD = 0.139) compared with rural areas (0.532, SD = 0.130; *p* = 0.003), and of the four WAMI components, two were significantly higher in urban areas than rural areas (A [assets] and M [maternal education]; Table 1). The mean W [water/sanitation] score was also numerically higher in urban areas (0.09, SD = 0.083) compared with rural areas (0.0089, SD = 0.066; *p* = 0.150).

The WAMI index was descriptively lower in the anti-HAV IgG-positive group (0.547, SD = 0.143) compared to the negative group (0.562, SD = 0.135). However, this was not statistically significant, and of the four WAMI components, the only one that showed a significant difference was the mother’s education (Table 1).

### 3.4. Knowledge of Hepatitis A and Hygienic Food Intake

A higher proportion of participants had heard of hepatitis A in urban areas (43.1%) compared to in rural areas (25.2%; *p* < 0.001; Table 4).

However, of those who had heard of the disease, only 45.3% in urban areas knew that it is communicable, compared to 60.2% in rural areas. Similarly, knowledge of contaminated food/water being the main cause of transmission was higher in rural areas (61.4%, compared to 34.9% in urban areas). Overall, 58.8% of participants knew that a vaccine to prevent hepatitis A was available in Indonesia. Across all participants in urban and rural areas, the most recognized symptom of hepatitis A was yellowish discoloration of the eyes (71.7%). An overall assessment of the participants’ knowledge of hepatitis A, calculated as a score based on their answers to the questionnaire, found that 65.5% of participants overall had an adequate understanding of hepatitis A, with a higher proportion in rural (74.7%) vs. urban areas (56.4%; *p* < 0.001). No significant differences in terms of adequate knowledge of HAV were found between the positive and negative anti-HAV IgG groups (Table 4).

In terms of hygiene, significant differences were found between urban and rural areas in the responses to questions relating to hygienic food preparation and intake (Table 5). Higher proportions of participants in rural areas prepared food on the ground (rural: 30.9% vs. urban: 14.1%), while lower proportions of participants in rural areas reported always washing their hands before handling food (rural: 51.8% vs. urban: 85.2%) and before eating food (rural: 77.5% vs. urban: 91.1%). There were significant differences between the anti-HAV IgG-seropositive and -seronegative groups in their reported behaviors in cooking main meals at home and washing hands before handling food and after defecation (Table 4).

### 3.5. Multivariate Analysis

The factors identified as associated with HAV IgG seropositivity (*p* < 0.20) were included in a multivariate model to show the simultaneous influence of the risk factors on the incidence of anti-HAV IgG seropositivity (Table 5).

The coefficient of determination (R^2^) of 54.6% shows that anti-HAV IgG seropositivity is influenced by living in a rural area, age, father’s education, and the cleanliness of the kitchen used to prepare food (Table 5). The remaining factors were not included in the multivariate regression model as the variables were not significant in the univariate models.

## 4. Discussion

In this cross-sectional study, we found that the seroprevalence of hepatitis A was higher in females compared to males, consistent with a previous study [22], and in those living in rural areas compared to urban areas, as reported previously in Vietnam and Indonesia [23]. However, other studies found a higher prevalence in males [24,25]. Low SES, high-density housing, and inadequate water treatment contributed to the endemicity. Previous studies have also shown that HAV seropositivity is strongly correlated with social factors, access to clean water, and improved sanitation [1,2], and worldwide, in general, urban areas have seen a decline in hepatitis A infection, whereas rates in rural areas remain high. Additionally, the prevalence is generally lower among higher social classes [14,15].

We found that approximately one-third of all participants had heard of hepatitis A, and over half of these knew that a vaccine to prevent hepatitis A was available in Indonesia. However, the vaccine is only available through private healthcare services and is not yet included in the national immunization program. Eight (0.6%) participants reported that they had previously been diagnosed with hepatitis, of whom seven were anti-HAV IgG positive. None of the participants had been vaccinated against hepatitis A (confirmed against their vaccination card).

A previous study found that a higher level of perceived knowledge was associated with better actual knowledge, better practices, and increased willingness to get vaccinated [26]. Overall, in both urban and rural areas, the participants’ understanding of hepatitis A was categorized as “sufficient”. The symptom of hepatitis A that participants in both urban and rural areas were most aware of is the change in eye color to yellowish, whilst significantly more participants in urban areas were aware of dark tea-colored urine and fever compared to those in rural areas. In rural areas, it is widely known that HAV is communicable and that contaminated food/water is the main method of transmission.

A high proportion of participants in this study reported washing their hands before handling food, which could be due to the promotion of hand washing in Indonesia during the COVID-19 pandemic. Although hand washing before eating was more common in urban than in rural areas, there was no significant difference in anti-HAV IgG seropositivity, whereas there were associations with other factors relating to the hygienic intake of food, such as washing hands before handling food and washing hands after defecation.

The youngest age at which the seroprevalence was equal to 50%, referred to as the AMPI, was lower in the rural area compared to the urban area (19.5 years for rural and 24.5 years for urban), resulting in an overall population AMPI of 22 years. These AMPI results are classified as intermediate endemicity (15–34 years). In the last study in Indonesia, conducted in 1996 in an urban area of Yogyakarta, low endemicity was reported [9]. Regionally and geographically, the sample size and research methods are similar to our study [9,10]; however, there are different study results due to the transition in endemicity with high variability in HAV seroprevalence across the country.

This study supports the evidence of the ongoing endemicity shift reported across several Southeast Asian countries that have experienced major socioeconomic improvements in the past two decades [9,16,27].

One implication of intermediate endemicity status is that a substantial proportion of adolescents and adults remain susceptible to infection, and HAV may circulate, often through regular community-wide outbreaks. HAV infection in adolescents and adults is associated with a higher rate of severe clinical manifestations [28]. Adults are more likely to develop signs and symptoms of the disease than children. The severity of the disease is higher in older age groups with manifestations such as fulminant hepatitis. Infected children under 6 years of age are usually asymptomatic, and only 10% develop jaundice [11].

Therefore, populations in middle-income countries such as Indonesia may benefit from large-scale HAV vaccination (national immunization programs). Many vaccines have been included in the national immunization program in Indonesia, such as hepatitis B, Polio, BCG, DPT, HIB, PCV, Rotavirus, MR, and HPV, but the hepatitis A vaccine has not been included [29]. 

Countries around Indonesia, such as Malaysia, Singapore, Thailand, and the Philippines, do not currently have data on hepatitis A vaccine coverage in children, because the vaccine is not yet included in their national immunization programs. HAV cases in Malaysia number 0.30 per 100,000 individuals, while those in Singapore number approximately 1.0 per 100,000 population; Thailand has an incidence rate of 0.65 per 100,000, and the Philippines ranges between 0.4 and 0.6 cases per 100,000 population [9].

This research was conducted in a community setting in two different geographic locations with a large age range with several limitations in this study.

In conclusion, our study found that the AMPI was within the 20–24-year age group, with an age point of 22 years, classified as intermediate HAV endemicity status. This shows a shift in HAV endemicity status from low to intermediate, supporting the need for large-scale national hepatitis A vaccination in Indonesia.

## 5. Conclusions

This study revealed a shift in hepatitis A endemicity in Indonesia from low to intermediate status, with an average midpoint of infection (AMPI) of 22 years. Rural areas showed a younger AMPI (19.5 years) compared to urban areas (24.5 years), highlighting the impact of socioeconomic factors, inadequate sanitation, and water treatment on HAV transmission. Although awareness of hepatitis A and its vaccine was limited, the absence of a national vaccination program underscores a public health gap.

Given the intermediate endemicity, with many adolescents and adults still susceptible to HAV, we advocate for the inclusion of hepatitis A vaccination in Indonesia’s national immunization program to reduce transmission and prevent severe clinical outcomes. Further research is needed to refine strategies for nationwide prevention.

## Figures and Tables

**Figure 1 medicina-61-00806-f001:**
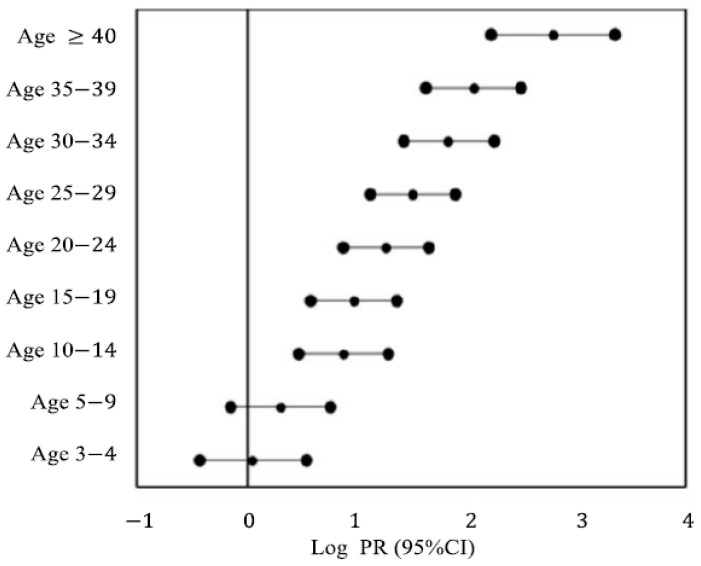
Forest plot for the total prevalence ratio (PR) of anti-HAV IgG positivity and 95% CI.

**Figure 2 medicina-61-00806-f002:**
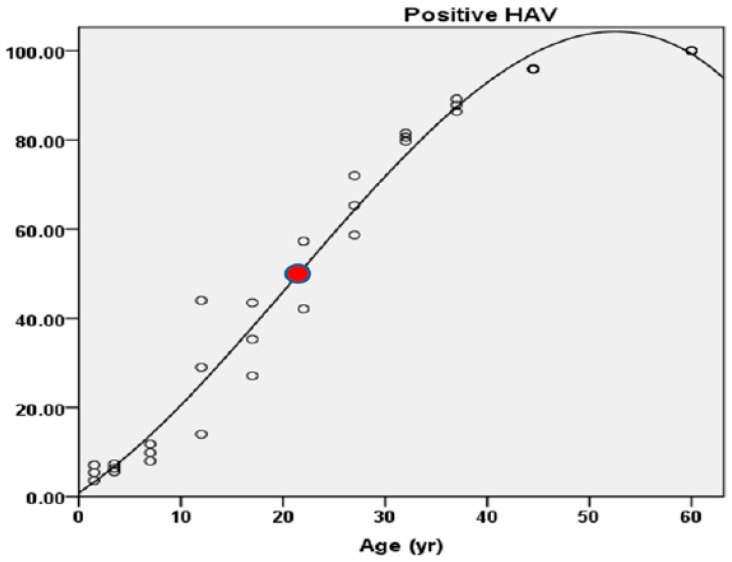
Relationship between age and percentage of HAV IgG positivity by age group with AMPI. Red dot indicates 50% positive HAV IgG.

**Figure 3 medicina-61-00806-f003:**
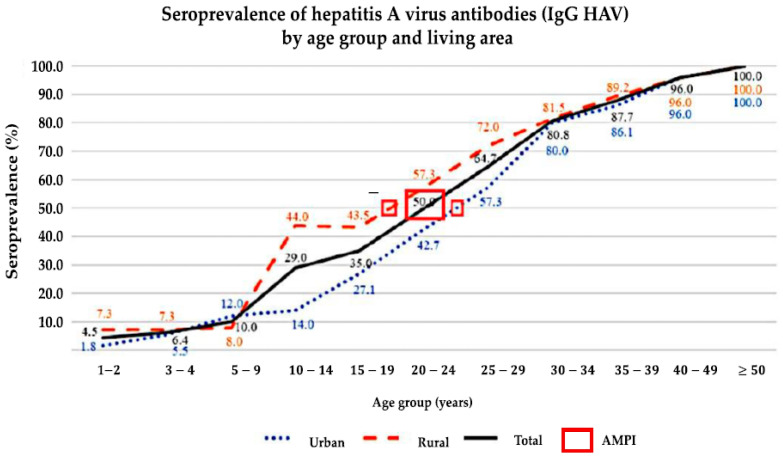
Seroprevalence trend of HAV IgG antibodies by age group and by living area.

**Table 1 medicina-61-00806-t001:** Sociodemographic characteristics of subjects per living area and anti-HAV IgG seroprevalence.

Characteristics	Total *n* = 1280	Living Area		*p* Value *	Anti-HAV IgG		*p* Values *
		Urban *n* = 640	Rural *n* = 640		Positive *n* = 647 (50.55%)	Negative *n* = 633 (49.45%)	
Sex:				0.001			0.001
Male	506 (39.5%)	283 (44.2%)	223 (34.8%)		227 (44.9%)	279 (55.1%)	
Female	774 (60.5%)	357 (55.8%)	417 (65.2%)		420 (54.3%)	354 (45.7%)	
Age (years):				1.00			<0.001
1–2	110 (8.6%)	55 (8.6%)	55 (8.6%)		5 (4.5%)	105 (95.5%)	
3–4	110 (8.6%)	55 (8.6%)	55 (8.6%)		7 (6.4%)	103 (93.6%)	
5–9	100 (7.8%)	50 (7.8%)	50 (7.8%)		10 (10.0%)	90 (90.0%)	
10–14	100 (7.8%)	50 (7.8%)	50 (7.8%)		29 (29.0%)	71 (71.0%)	
15–19	140 (10.9%)	70 (10.9%)	70 (10.9%)		49 (35.0%)	91 (65.0%)	
20–24	150 (11.7%)	75 (11.7%)	75 (11.7%)		75 (50.0%)	75 (50.0%)	
25–29	150 (11.7%)	75 (11.7%)	75 (11.7%)		97 (64.7%)	53(35.3%)	
30–34	130 (10.2%)	65 (10.2%)	65 (10.2%)		105 (80.8%)	25 (19.2%)	
35–39	130 (10.2%)	65 (10.2%)	65 (10.2%)		114 (87.7%)	16 (12.3%)	
40–49	100 (7.8%)	50 (7.8%)	50 (7.8%)		96 (96.0%)	4 (4.0%)	
≥50	60 (4.7%)	30 (4.7%)	30 (4.7%)		60 (100%)	0 (0%)	
Education of father:				<0.001			<0.001
Illiterate	9 (0.7%)	6 (0.9%)	3 (0.5%)		9 (100%)	0 (0%)	
Primary school (6 years)	382 (29.9%)	182 (28.5%)	200 (31.3%)		277 (72.5%)	105 (27.5%)	
Middle school (9 years)	274 (21.5%)	114 (17.8%)	160 (25.1%)		143 (52.2%)	131 (47.8%)	
High school (13 years)	531 (41.6%)	278 (43.5%)	253 (39.7%)		188 (35.4%)	343 (64.6%)	
Graduate/post-graduate (16+ years)	81 (6.3%)	59 (9.2%)	22 (3.4%)		27 (33.3%)	54 (66.7%)	
Education of mother:				<0.001			<0.001
Illiterate	7 (0.5%)	6 (0.9%)	1 (0.2%)		7 (100%)	0 (0%)	
Primary school (6 years)	428 (33.5%)	191 (29.9%)	237 (37.1%)		308 (72.0%)	120 (28.0%)	
Middle school (9 years)	310 (24.3%)	147 (23.0%)	163 (25.5%)		155 (50.0%)	155 (50.0%)	
High school (13 years)	468 (36.6%)	246 (38.5%)	222 (34.7%)		152 (32.5%)	316 (67.5%)	
Graduate/post-graduate (16+ years)	65 (5.1%)	49 (7.7%)	16 (2.5%)		23 (35.4%)	42 (64.6%)	
Occupation:				0.004			<0.001
Professional	69 (5.4%)	41 (6.4%)	28 (4.4%)		48 (69.6%)	21 (30.4%)	
Semi-professional	3 (0.2%)	1 (0.2%)	2 (0.3%)		3 (100%)	0 (0%)	
Clerical/shop owner	37 (2.9%)	22 (3.4%)	15 (2.3%)		27 (73.0%)	10 (27.0%)	
Skilled worker	66 (5.2%)	34 (5.3%)	32 (5.0%)		44 (66.7%)	22 (33.3%)	
Semi-skilled worker	35 (2.7%)	19 (3.0%)	16 (2.5%)		23 (65.7%)	12 (34.3%)	
Unskilled worker	103 (8.1%)	66 (10.3%)	37 (5.8%)		86 (83.6%)	17 (16.5%)	
Household duties	319 (24.9%)	133 (20.8%)	186 (29.1%)		262 (82.1%)	57 (17.9%)	
Unemployed (adults)	73 (5.7%)	36 (5.6%)	37 (5.8%)		39 (53.4%)	34 (46.6%)	
Attending school	318 (24.9%)	158 (24.7%)	160 (25.0%)		93 (29.2%)	225 (70.8%)	
Attending garderie/pre-school	40 (3.1%)	23 (3.6%)	17 (2.7%)		4 (10.0%)	36 (90.0%)	
At home (children)	208 (16.3%)	99 (15.5%)	109 (17.0%)		12 (5.8%)	196 (94.2%)	
Other	8 (0.6%)	7 (1.1%)	1 (0.2%)		6 (75.0%)	2 (25.0%)	
Total number of family members living in the same household:				0.029			0.001
<2	21 (1.6%)	11 (1.7%)	10 (1.6%)		18 (85.7%)	3 (14.3%)	
2–3	241 (18.9%)	107 (16.7%)	134 (21.0%)		140 (57.6%)	103 (42.4%)	
4–5	683 (53.4%)	337 (52.7%)	346 (54.2%)		334 (48.9%)	349 (51.1%)	
6–9	291 (22.8%)	156 (24.4%)	135 (21.2%)		138 (57.3%)	153 (52.6%)	
≥10	42 (3.3%)	29 (4.5%)	13 (2.0%)		17 (40.5%)	25 (59.5%)	
The number of rooms in house:				0.058			0.800
<2	270 (21.1%)	140 (21.9%)	133 (20.8%)		138 (51.1%)	132 (48.9%)	
2–3	758 (59.4%)	358 (56.0%)	400 (62.5%)		385 (50.8%)	373 (49.2%)	
4–5	216 (16.9%)	122 (19.0%)	94 (14.7%)		107 (49.5%)	109 (50.5%)	
≥5	33 (2.6%)	20 (3.1%)	13 (2.0%)		14 (42.4%)	19 (57.6%)	
WAMI:							
W score		0.095 (0.083)	0.089 (0.066)	0.15	0.100 (0.087) *	0.090 (0.079) *	0.097
A score		0.141 (0.048)	0.130 (0.053)	<0.001	0.142 (0.049) *	0.139 (0.048) *	0.47
M score		0.152 (0.050)	0.142 (0.044)	<0.001	0.133 (0.050) *	0.168 (0.044) *	<0.001
I score		0.168 (0.055)	0.171 (0.051)	0.217	0.171 (0.052) *	0.165 (0.057) *	0.1.61
WAMI index:				0.003			0.164
Mean (SD)		0.555 (0.139)	0.532 (0.130)		0.547 (0.143) *	0.562 (0.135) *	
Median		0.562	0.547		0.547	0.562	
Range		0.219–0.969	0.188–0.906		0.234–0.938	0.219–0.969	
0.188–0.344		50 (7.8%)	67 (10.5%)		24 (8.1%)	26 (7.6%)	
>0.344–0.500		202 (31.6%)	209 (32.7%)		103 (34.7%)	99 (28.9%)	
>0.500–0.656		219 (34.2%)	230 (35.9%)		94 (31.6%)	125 (36.4%)	
>0.656–0.813		149 (23.3%)	131 (20.5%)		67 (22.6%)	82 (23.9%)	
>0.813–0.969		20 (3.1%)	3 (0.5%)		9 (3.0%)	11 (3.2%)	
The main source of drinking water for members of the household:				<0.001			<0.001
a. Piped water into dwelling (i)		100 (15.6%)	22 (3.4%)		63 (9.7%)	59 (9.3%)	
b. Tubewell/bore-hole (i)		69 (10.8%)	9 (1.4%)		44 (6.8%)	34 (5.4%)	
c. Protected dug well (i)		8 (1.3%)	3 (0.5%)		3 (0.5%)	8 (1.3%)	
d. Unprotected dug well (un)		2 (0.3%)	0		0	2 (0.3%)	
e. Protected spring (i)		8 (1.3%)	0		4 (0.6%)	4 (0.6%)	
f. Unprotected spring (un)		2 (0.3%)	0		1 (0.2%)	1 (0.2%)	
g. Bottled water (un)		451 (70.5%)	361 (56.4%)		441 (68.2%)	371 (58.6%)	
h. Cart with small tank/drum (un)		0	6 (0.9%)		5 (0.8%)	1 (0.2%)	
i. Other: refill gallon (un)		0	239 (37.3%)		86 (13.3%)	153 (24.2%)	
Source of drinking water:				0.003			0.624
Improved		185 (28.9%)	34 (5.3%)		114 (17.6%)	105 (16.6%)	
Unimproved		455 (71.1%)	606 (94.7%)		533 (82.4%)	528 (83.4.%)	
Sanitation: If using a “flush” or “pour flush” probe, where does the waste go?				<0.001			0.010
a. Piped sewer system (i)	56 (4.4%)	11 (1.7%)	45 (7.0%)		36 (5.6%)	20 (3.2%)	
b. Septic tank (i)	659 (51.5%)	289 (45.2%)	370 (57.8%)		351 (54.3%)	308 (48.7%)	
c. Pit latrine (i)	5 (0.4%)	0	5 (0.8%)		4 (0.6%)	1 (0.2%)	
d. Elsewhere (un)	30 (2.3%)	24 (3.8%)	6 (0.9%)		12 (1.9%)	18 (2.8%)	
e. No facilities or bush or field (un)	2 (0.2%)	0	2 (0.3%)		2 (0.3%)	0	
f. Other: ditch (river) (un)	521 (40.7%)	314 (49.1%)	207 (32.3%)		240 (37.1%)	281 (44.4%)	
g. Do not know (un)	7 (0.5%)	2 (0.3%)	5 (0.8%)		2 (0.3%)	5 (0.8%)	
Sanitation facility:				<0.001			0.002
Improved	720 (56.25%)	300 (46.9)	420 (65.6)		391 (60.4)	329 (52.0)	
Unimproved	560 (43.75%)	340 (53.1	220 (34.4)		256 (39.6)	304 (48.0)	

i: improved; un: unimproved; * Chi-square test to compare urban vs. rural and anti-HAV IgG positive vs. negative.

**Table 2 medicina-61-00806-t002:** Anti-HAV IgG in urban vs. rural living areas and anti-HAV IgG positive vs. negative results.

Anti-HAV IgG (S/CO)	Living Area		*p* Value	Anti-HAV IgG		Total (S/CO)
	Urban *n* = 640	Rural *n* = 640		Positive *n* = 647	Negative *n* = 633	
Average (SD)	4.75 (5.048)	5.73 (5.29)	<0.001 *	10.192 (1.953)	0.186 (0.121)	5.24 (5.19)
Median	0.32	7.89		10.62	0.15	1.96
Range	0.07–13.06	0.05–19.29		1.01–19.29	0.05 –0.95	0.05–19.29
Seroprotected			0.003 **			
Positive	297 (46.4%)	350 (54.7%)				647 (50.6%)
Negative	343 (53.6%)	290 (45.3%)				633 (49.4%)

HAV: hepatitis A virus; S/CO: signal-to-cut-off ratio; * Mann–Whitney test; ** Uji Chi-square. Positive if anti-HAV IgG ≥ 1 S/CO.

**Table 3 medicina-61-00806-t003:** Seroprevalence of hepatitis A based on Anti-HAV IgG, by age group and living area.

Characteristics	Urban		*p* Value *	Rural		*p* Value **	Positivity Between Urban and Rural *p* Value ***
	Positive *n* (%) 297 (46.4)	Negative *n* (%) 343 (53.6)		Positive *n* (%) 350 (54.7)	Negative *n* (%) 290 (45.3)		
Sex			0.049			0.020	
Male	119 (42.0%)	164 (58.0%)		108 (48.4%)	115 (51.6%)		0.152
Female	178 (49.9%)	179 (50.1%)		242 (58.0%)	175 (42.0%)		0.023
Age (Years)			<0.001 *			<0.001 *	
1–2	1 (1.8%)	54 (98.2%)		4 (7.3%)	51 (92.7%)		0.363
3–4	3 (5.5%)	52 (94.5%)		4 (7.3%)	51 (92.7%)		1.000
5–9	6 (12.0%)	44 (88.0%)		4 (8.0%)	46 (92.0%)		0.741
10–14	7 (14.0%)	43 (86.0%)		22 (44.0%)	28 (56.0%)		0.001
15–19	19 (27.1%)	51 (72.9%)		30 (42.9%)	40 (57.1%)		0.051
20–24	32 (42.7%)	43 (57.3%)		43 (57.3%)	32 (42.7%)		0.072
25–29	43 (57.3%)	32 (42.7%)		54 (72.0%)	21 (28.0%)		0.060
30–34	52 (80.0%)	13 (20.0%)		53 (81.5%)	12 (18.5%)		0.824
35–39	56 (86.1%)	9 (13.9%)		58 (89.2%)	7 (10.8%)		0.593
40–49	48 (96.0%)	2 (4.0%)		48 (96.0%)	2 (4.0%)		1.000
≥50	30 (100%)	0		30 (100%)	0		1.000

HAV: hepatitis A virus; * Chi-square test or Fisher exact test for expectation cell < 5, *p*-value * positive vs. negative in urban area, *p*-value ** positive vs. negative in rural area, and *p*-value *** for positivity difference between urban and rural areas.

**Table 4 medicina-61-00806-t004:** Association of knowledge of disease and hygienic food intake with living area and anti-HAV IgG serostatus.

Variables	Total	Living Area		*p* Value	Anti-HAV Ig G		*p* Value
		Urban	Rural		Positive	Negative	
Knowledge of disease							
1. Heard about a disease called hepatitis A before (yes) Kind of disease:	437 (34.1%)	276 (43.1%)	161 (25.2%)		217 (33.5%)	220 (34.8%)	
a. Communicable	222 (50.8%)	125 (45.3%)	97 (60.2%)	<0.001	117 (53.4%)	105 (47.3%)	0.647
b. Non-communicable	102 (23.5%)	64 (23.2%)	38 (23.6%)	0.001	47 (21.55)	55 (24.8%)	0.433
c. Do not know	113 (25.8%)	87 (31.5%)	26 (16.1%)		55 (25.1%)	62 (27.9%)	
2. The main way of transmission				<0.001			0.009
a. By blood	71 (20.5%)	57 (24.6%)	14 (12.3%)		24 (14.4%)	47 (26.3%)	
b. By air	55 (15.9%)	46 (19.8%)	9 (7.9%)		21 (12.6%)	34 (19.0%)	
c. Sexually transmitted	14 (4.0%)	13 (5.6%)	1 (0.9%)		6 (3.6%)	8 (4.5%)	
d. By contaminated food/water	151 (43.6%)	81 (34.9%)	70 (61.4%)		81 (48.5%)	70 (39.1%)	
e. By mosquito bite	17 (4.9%)	16 (6.9%)	1 (0.9%)		11 (6.6%)	6 (3.4%)	
f. Combination	38 (11.0%)	19 (8.2%)	19 (16.7%)		24 (14.4%)	14 (7.8%)	
3. A vaccine to prevent hepatitis A available in Indonesia							
a. Yes	257 (58.8%)	165 (59.8%)	92 (57.1%)	0.522	128 (59.3%)	129 (58.4%)	0.494
b. No	35 (8.0%)	19 (6.9%)	16 (9.9%)		14 (6.5%)	21 (9.5%)	
c. Do not know	145 (33.2%)	92 (33.3%)	53 (32.9%)		74 (34.3%)	71 (32.1%)	
4. The possible risk factor/factors	(*n* = 439)	(*n* = 279)	(*n* = 160)				
a. Using unclean toilets	191 (43.5%)	125 (44.8%)	66 (41.2%)	0.470	97 (15.0%)	94 (14.8%)	0.943
b. Consuming contaminated water/food	193 (44.0%)	100 (35.8%)	93 (58.1%)	<0.001	93 (14.4%)	99 (15.6%)	0.526
c. Talking to someone who are hepatitis A infected	60 (13.7%)	30 (10.8%)	30 (18.8%)	0.019	28 (4.3%)	32 (5.0%)	0.538
d. Sharing a room with an infected individual	70 (15.9%)	45 (16.1%)	25 (15.6%)	0.890	33 (5.1%)	37 (5.8%)	0.558
e. Do not know	108 (24.6%)	86 (30.8%)	22 (13.8%)	<0.001	55 (8.5%)	53 (8.4%)	0.934
5. The possible symptoms of hepatitis A	(*n* = 442)	(*n* = 280)	(*n* = 162)				
a. Yellowish discoloration of eyes	317 (71.7%)	193 (68.9%)	124 (76.5%)	0.087	165 (75.0%)	152 (68.5%)	0.029
b. Abdominal pain	188 (42.5%)	118 (42.1%)	70 (43.2%)	0.827	93 (42.3%)	95 (42.8%)	0.078
c. Nasal bleeding	78 (17.6%)	53 (18.9%)	25 (15.4%)	0.353	37 (16.8%)	41 (18.5%)	0.737
d. Dark tea-colored urine	195 (44.1%)	134 (47.9%)	61 (37.7%)	0.038	98 (44.5%)	97 (43.7%)	0.981
e. Numbness in extremities	83 (18.8%)	52 (18.6%)	31 (19.1%)	0.884	40 (18.2%)	43 (19.5%)	0.750
f. Fever	261 (59.0%)	179 (63.9%)	82 (50.6%)	0.006	130 (59.1%)	131 (59.0%)	0.618
g. Pale stools	146 (33.0%)	96 (34.3%)	50 (30.9%)	0.462	69 (31.4%)	77 (34.7%)	0.678
6. Knowledge of HAV **				<0.001			0.645 *
Adequate	839 (65.5%)	361 (56.4%)	478 (74.7%)		428 (66.2%)	411 (64.9%)	
Inadequate	441 (34.5%)	279 (43.6%)	162 (25.3%)		219 (33.8%)	222 (35.1%)	
Hygienic food intake							
1. Prepare food at home				<0.001			0.278
a. On the ground	288 (22.5%)	90 (14.1%)	198 (30.9%)		157 (24.3%)	131 (20.7%)	
b. Multi-purpose table	447 (34.9%)	203 (31.6%)	245 (38.3%)		217 (33.5%)	230 (36.4%)	
c. Table exclusively set for cooking	544 (42.5%)	347 (54.3%)	197 (30.8%)		273 (42.2%)	271 (42.9%)	
2. Main meals from home				<0.001			0.004
a. Never	22 (1.7%)	10 (1.6%)	12 (1.9%)		15 (2.3%)	7 (1.1%)	
b. Sometimes	300 (23.4%)	115 (18.0%)	185 (28.9%)		168 (26.0%)	132 (20.9%)	
c. Most of the time	194 (15.2%)	157 (24.5%)	37 (5.8%)		108 (16.7%)	86 (13.6%)	
d. Always	764 (59.7%)	358 (55.9%)	406 (63.4%)		356 (55.0%)	408 (64.5%)	
3. Main meals from street				<0.001			0.075
a. Never	393 (30.7%)	208 (32.5%)	185 (28.9%)		206 (31.8%)	187 (29.5%)	
b. Sometimes	760 (59.4%)	349 (54.5%)	411 (64.2%)		365 (56.4%)	395 (62.4%)	
c. Most of the time	98 (7.7%)	68 (10.6%)	30 (4.7%)		59 (9.1%)	39 (6.2%)	
d. Always	29 (2.3%)	15 (2.3%)	14 (2.2%)		17 (2.6%)	12 (1.9%)	
4. Wash hands before handling food				<0.001			0.032
a. Never	15 (1.2%)	3 (0.5%)	12 (1.9%)		13 (2.0%)	2 (0.3%)	
b. Sometimes	210 (16.4%)	64 (10.0%)	146 (22.8%)		109 (16.9%)	101 (16.0%)	
c. Most of the time	178 (13.9%)	28 (4.4%)	150 (23.5%)		93 (14.4%)	85 (13.4%)	
d. Always	876 (68.5%)	545 (85.2%)	331 (51.8%)		431 (66.7%)	445 (70.3%)	
5. Wash hands before eating food				<0.001			0.402
a. Never	3 (0.2%)	0	3 (0.5%)		2 (0.3%)	1 (0.2%)	
b. Sometimes	75 (5.9%)	4 (0.6%)	38 (5.9%)		32 (4.9%)	43 (6.8%)	
c. Most of the time	123 (9.6%)	8 (1.3%)	103 (16.1%)		67 (10.4%)	56 (8.8%)	
d. Always	1079 (84.3%)	627 (98.1%)	496 (77.5%)		546 (84.4%)	533 (84.2%)	
6. Wash hands after defecation (in case of younger children)				0.024			0.006
a. Never	1 (0.1%)	0	1 (0.2%)		0	1 (0.2%)	
b. Sometimes	21 (1.6%)	4 (0.6%)	17 (2.7%)		3 (0.5%)	18 (2.8%)	
c. Most of the time	14 (1.1%)	8 (1.3%)	6 (0.9%)		7 (1.1%)	7 (1.1%)	
d. Always	1242 (97.2%)	627 (98.1%)	615 (96.2%)		636 (98.5%)	606 (95.9%)	
7. The kitchen which prepared the food: free of insects and rodents				<0.001			0.077
a. Never	392 (30.6%)	148 (23.1%)	244 (38.1%)		207 (32.0%)	185 (29.2%)	
b. Sometimes	686 (53.6%)	391 (61.1%)	295 (46.1%)		326 (50.4%)	360 (56.9%)	
c. Most of the time	85 (6.6%)	40 (6.3%)	45 (7.0%)		51 (7.9%)	34 (5.4%)	
d. Always	117 (9.1%)	61 (9.5%)	56 (8.8%)		63 (9.7%)	54 (8.5%)	

HAV: hepatitis A virus; * Chi-square test; ** Knowledge about HAV disease: if items 1 to 5 are answered yes, score 1 (adequate); others are scored 0 (inadequate).

**Table 5 medicina-61-00806-t005:** Multiple logistic regression analysis of factors related to anti-HAV IgG positivity: forward LR method.

Variables	Coeff B	SE (B)	*p*-Value	PR*_adj_* (95% CI)
Living area (rural vs. urban)	0.485	0.154	0.002	1.62 (1.20–2.19)
Age (years) *				
3–4	0.124	0.577	0.830	1.13 (0.37–3.50)
5–9	0.705	0.539	0.198	2.02 (0.70–5.82)
10–14	2.029	0.480	<0.001	7.60 (2.97–19.47)
15–19	2.247	0.459	<0.001	9.46 (3.84–23.26)
20–24	2.927	0.454	<0.001	18.68 (7.67–45.46)
25–29	3.477	0.459	<0.001	32.35 (13.16–79.57)
30–34	4.254	0.483	<0.001	69.75 (27.08–179.63)
35–39	4.760	0.508	<0.001	116.72 (43.12–315.90)
≥40	6.433	0.666	<0.001	621.97 (168.77–2292.18)
Education of father **				
Illiterate and primary school (6 years)	0.673	0.327	0.040	1.96 (1.03–3.72)
Middle school (9 years)	0.716	0.332	0.031	2.05 (1.07–3.92)
High school (13 years)	0.254	0.311	0.414	1.29 (0.70–2.37)
The kitchen which prepared the food: free of insects and rodents	0.596	0.217	0.006	1.82 (1.19–2.78)

Accuracy = 79.4%; R^2^ (Nagelkerke) = 54.6%. PR (95% CI): prevalence ratio (95% confidence interval); R^2^: R-Squared or the coefficient of determination; * reference age 1–2; ** reference graduate/post-graduate.

## Data Availability

The data supporting the reported results in this study are available upon reasonable request from the corresponding author. The data are not publicly available due to privacy restrictions.

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
