# Peer review of "Seroprevalence and Shifting Endemicities of Hepatitis A Virus Infection in Two Contrasting Geographical Areas in Indonesia"

_medicina, 2025, doi:10.3390/medicina61050806_

Round 1
Reviewer 1 Report
Comments and Suggestions for Authors
The authors analyzed the seroprevalence of HAV infection (exposure) in an urban and a rural area from Indonesia. They found an intermediate prevalence, and suggest that a shift from low to intermediate endemicity has occurred. They propose vaccination based on this shift. Some concerns should be addressed before acceptance of this manuscript.
- The authors found a higher seroprevalence in women compared to men, and this result is somehow unexpected (see Choe YJ and Son H, Vaccine 2020, 38, 712-714). The authors should discuss better this observation.
- HAV acquisition at higher age may more symptomatic, and if combined with previous liver conditions (such as HCV infection) may lead to fulminant hepatitis. This should be discussed in this study. Some of this information is mentioned in line 291, but might be more extended.
- The observation of shifting (increasing) endemicity is based on a single report of low endemicity in Yogyakarta. The difference in endemicity may be also due to regional differences. In addition, both studies analyzed 1200 and 1100 subjects, and some limitations may hamper the validity of this conclusion: the sample size, differences in testing methodologies. This should be addressed with more details.
Author Response
- The authors found a higher seroprevalence in women compared to men, and this result is somehow unexpected (see Choe YJ and Son H, Vaccine 2020, 38, 712-714). The authors should discuss better this observation.
Response: We have revised with additional results suggested by the reviewer:
We found that the seroprevalence of hepatitis A was highest in females compared to males, consistent with a previous study. However, other studies show that males are predominant.
add at line 245
2. HAV acquisition at higher age may more symptomatic, and if combined with previous liver conditions (such as HCV infection) may lead to fulminant hepatitis. This should be discussed in this study. Some of this information is mentioned in line 291, but might be more extended.
Response we agree to add
Adults are more likely to develop signs and symptoms of the disease than children. The severity of the disease is higher in older age groups with manifestations such as fulminant hepatitis. Infected children under 6 years of age are usually asymptomatic and only 10% develop jaundice. line 288-291
3. The observation of shifting (increasing) endemicity is based on a single report of low endemicity in Yogyakarta. The difference in endemicity may be also due to regional differences. In addition, both studies analyzed 1200 and 1100 subjects, and some limitations may hamper the validity of this conclusion: the sample size, differences in testing methodologies. This should be addressed with more details.
The response we add according to the reviewer's suggestion:
In the last study in Indonesia, from 1996 in an urban area of Yogyakarta, low endemicity was reported [9]. The study showed that regionally, and geographically, the number of samples and research methods were similar, thus providing a basis for comparison, contrasting with the results of this study and highlighting either the transition in endemicity or high variability in HAV seroprevalence across the country [9,10].
Add lines 278-279

Reviewer 2 Report
Comments and Suggestions for Authors The authors of the manuscript present data from an observational, cross-sectional study of hepatitis A seroprevalence in two contrasting geographic areas in Indonesia. The research design is well structured to achieve the stated objectives.Analyzes of the obtained results show a change in hepatitis A endemicity in Indonesia from
low to medium status. Based on the obtained results, the authors of the manuscript recommend the inclusion of the hepatitis A vaccine in the national immunization program The study has its positive aspects with a view to enriching the data on the serological prevalence of hepatitis A in middle-income countriesc. My remarks are on the text from lines 118 to 131. I consider that these are recommendations that are automatically inserted into the text.
Author Response
The authors of the manuscript present data from an observational, cross-sectional study of hepatitis A seroprevalence in two contrasting geographic areas in Indonesia. The research design is well structured to achieve the stated objectives.
Analyzes of the obtained results show a change in hepatitis A endemicity in Indonesia from
low to medium status. Based on the obtained results, the authors of the manuscript recommend the inclusion of the hepatitis A vaccine in the national immunization program The study has its positive aspects with a view to enriching the data on the serological prevalence of hepatitis A in middle-income countries.
My remarks are on the text from lines 118 to 131. I consider that these are recommendations that are automatically inserted into the text.
Respons
Thank you for your suggestion:
The Automatically templet we have to delete

Reviewer 3 Report
Comments and Suggestions for Authors
1. In the introduction, please indicate that class G antibodies appear 8-12 weeks after infection and persist for life
2. In the introduction, it is necessary to explain what geographical differences there are between "Two Contrasting Geographical Areas in Indonesia" (temperature? highlands vs. lowlands? differences in rainfall, presence vs. absence of vegetation? etc.)
3. Lines 117-131 contain information from the manuscript template and should be deleted.
4. Line 223: Hygiene - with a lowercase letter.
5. In the Discussion, please provide examples of countries in your region where all children are vaccinated and the statistics of HAV cases in these countries.
6. If you propose to introduce vaccination into the national program, please explain which vaccines are already included in the national immunization program in Indonesia. Please explain what benefits this has.
Author Response
Thank you for your input and suggestions
- In the introduction, please indicate that class G antibodies appear 8-12 weeks after infection and persist for life
Response:
The body produces Immunoglobulin M (IgM) antibodies when first exposed to hepatitis A. These antibodies will remain in the blood for about 2-12 months. Immunoglobulin G (IgG) antibodies appear after being infected with the Hepatitis A virus. These antibodies appear 8 to 12 weeks after infection or vaccination and will remain in the blood and protect against hepatitis A permanently.
add lines 44-48
- In the introduction, it is necessary to explain what geographical differences there are between "Two Contrasting Geographical Areas in Indonesia" (temperature? highlands vs. lowlands? differences in rainfall, presence vs. absence of vegetation? etc.)
Response:
Two Contrasting Geographical Areas in Indonesia between urban areas (lowlands) that have an altitude of 768 meters above sea level (asl) with an average temperature of 23.5 °C and an average rainfall of around 200.4 mm per month, with vegetation varying from low to high greenery, while rural areas (highlands) have an altitude ranging from 750 to 1,200 meters above sea level (asl) with an average monthly temperature of 23.1 °C and rainfall of around 250 mm per month with high-density vegetation.
Add line 61-67
- Lines 117-131 contain information from the manuscript template and should be deleted.
Response:
Automatically template was delate
- Line 223: Hygiene - with a lowercase letter.
Response:
Hygiene change to hygiene
At line 215
- In the Discussion, please provide examples of countries in your region where all children are vaccinated and the statistics of HAV cases in these countries.
Response
Countries around Indonesia such as Malaysia, Singapore, Thailand, Philippines until now do not have data on hepatitis A vaccine coverage in children, because it is not yet included in the national immunization program. HAV cases in Malaysia: 0.30 per 100,000 individuals, Singapore approximately 1.0 per 100,000 population, Thailand incidence rate of 0.65 per 100,000, Philippines ranged between 0.4 and 0.6 cases per 100,000 population.
Add at lines 297-301
- If you propose to introduce vaccination into the national program, please explain which vaccines are already included in the national immunization program in Indonesia. Please explain what benefits this has.
Response:
Many vaccines have been included in the national immunization program in Indonesia, such as Hepatitis B, Polio, BCG, DPT, HIB, PCV, Rotavirus, MR, and HPV, but until now the Hepatitis A vaccine has not been included.
Line 293-295

Round 2
Reviewer 1 Report
Comments and Suggestions for Authors The authors addressed satisfactorely the concerns.
Just a small comment: Please rephrase this sentence in lines 268-269:
However, there are other studies that show that males are predominant [21, 22].
by
However, there are other studies found a higher prevalence in males [21, 22].
Author Response
Thank you for your review and suggestions
Reviewer 1
The authors addressed satisfactorely the concerns.
Just a small comment: Please rephrase this sentence in lines 268-269:
However, there are other studies that show that males are predominant [21, 22].
by
However, there are other studies found a higher prevalence in males [21, 22].
Comment:
We change as reviewer suggestion with
However, there are other studies found a higher prevalence in males [21, 22].
At line 245
